

# Source contributions of sulfur and nitrogen deposition – an HTAP II multi-model study on hemispheric transport

Jiani Tan[1], Joshua S. Fu[1], Frank Dentener[2], Jian Sun[1], Louisa Emmons[3], Simone Tilmes[3],

Johannes Flemming[4], Toshihiko Takemura[5], Huisheng Bian[6], Qingzhao Zhu[1], Cheng-En Yang[1],

Terry Keating[7]

[1] Department of Civil and Environmental Engineering, University of Tennessee, Knoxville, TN,
USA
[2] European Commission, Institute for Environment and Sustainability Joint Research Centre,
Ispra, Italy
[3] Atmospheric Chemistry Observations and Modeling Laboratory, National Center for
Atmospheric Research, Boulder, Colorado, USA
[4] Norwegian Meteorological Institute, Oslo, Norway
[5] Research Institute for Applied Mechanics, Kyushu University, Fukuoka, Japan
[6] National Aeronautics and Space Administration Goddard Space Flight Center, Greenbelt, MD,
USA
[7] US Environmental Protection Agency, Washington, DC, USA

*Correspondence to:* Joshua S. Fu (jsfu@utk.edu)

**Abstract.** With rising emissions by human activities, enhanced concentrations of air pollutants have been detected in hemispheric air flows in recent years, aggravating the regional air pollution and deposition burden. However, contributions of hemispheric air pollution to deposition at global scale have been given little attention in the literature. In this light, we assess the impact of hemispheric transport on sulfur (S) and nitrogen (N) deposition for 6 world regions: North America, Europe, South Asia, East Asia, Middle East and Russia in 2010, by using the multi-model ensemble results from the 2nd phase of Task Force Hemispheric Transport of Air Pollution (HTAP II) with and without 20% emission perturbation experiments. About 27-58%, 26-46% and 12-23% of local S, $NO_x$ and $NH_3$ emissions are transported and removed by deposition outside of the source regions annually, with 5% higher fraction of export in winter and 5% lower in summer. For receptor regions, 20% emission reduction in source regions affects the deposition in receptor regions by 1-10% for continental non-coastal regions and 1-15% for coastal regions and open oceans. Significant influences are found from North America to the North Atlantic Ocean (5-15%), from South Asia to western East Asia (2-10%) and from East Asia to the North Pacific Ocean (5-10%) and western North America (5-8%). The impact on



deposition caused by transport between neighbouring regions (i.e. Europe and Russia) occurs
throughout the whole year (slightly stronger in winter), while that by transport over long
distances (i.e. from East Asia to North America) mainly takes place in spring and fall, which is
consistent with the seasonality found for hemispheric transport of air pollutants. Deposition in
emission intense regions such as East Asia is dominated (~80%) by own region emission, while
deposition in low emission regions such as Russia is almost equally affected by own region
emission (~40%) and foreign impact (~23-45%). We also find that deposition on the coastal
regions or near coastal open ocean is twice more sensitive to hemispheric transport than non-
coastal continental regions, especially for regions (i.e. west coast of North America) in the
downwind location of major emission source regions. This study highlights the significant
impact of hemispheric transport on deposition in coastal regions, open ocean and low emission
regions. Further research is proposed for improving ecosystem and human health in these
regions, with regards to the enhanced hemispheric transport.

## 1 Introduction

The increasing consumption of energy by human activities has largely increased the deposition
of sulfur (S) and nitrogen (N) over the terrestrial and marine ecosystem in recent years (Kim et
al., 2011; Galloway et al., 2008; Duce et al., 2008) and the amount of atmospheric deposition is
estimated to continue increasing in the near future (Bleeker et al., 2011; Lamarque et al., 2013;
Kanakidou et al., 2016; Paulot et al., 2013; Lamarque et al., 2005). While deposition supplies
ecosystem with nutrients, too much deposition could cause various adverse impacts on the
environment, including acidification and eutrophication of the forest and waterbody (Bouwman
et al., 2002; Bergstrom and Jansson, 2006; Dentener et al., 2006; Phoenix et al., 2006), soil
acidification that slows down the crop production (Guo et al., 2010; Janssens et al., 2010) and
even decrease plant biodiversity (Bobbink et al., 2010; Clark and Tilman, 2008). The control of
deposition has become a growing worldwide concern.
Hemispheric transport of air pollutants is found to aggravate the regional air pollution
(Wild and Akimoto, 2001; Sudo and Akimoto, 2007; Fu et al., 2012; Fiore et al., 2009) as well as
enlarge the local deposition burden (Glotfelty et al., 2014; Sanderson et al., 2008). Air pollution
from Asia contributes to the concentration of $PM_{2.5}$ in western United States by 1.5 µg m$^{-3}$ (Tao
et al., 2016), the $O_3$ concentration by 3-10 ppbv (Zhang et al., 2009; Zhang et al., 2008; Yienger



et al., 2000; Reidmiller et al., 2009; Jacob et al., 1999; Brown-Steiner and Hess, 2011) and the peroxyacyl nitrate (PAN) concentration by 26 ppbv (Berntsen et al., 1999) in spring. The long-range transport of air pollution from North America is estimated to contribute by 3-5 ppb to (7-11%) to the $O_3$ concentration in Europe annually (Auvray and Bey, 2005; Guerova et al., 2006; Derwent et al., 2004; Li et al., 2002) and the increment can reach 25-28 ppbv during particular events (Guerova et al., 2006). European outflow affects the surface $O_3$ concentration in western China by 2-6 ppbv in spring and summer (Li et al., 2014) and North Africa by 5-20 ppbv in summer (Duncan et al., 2008; Duncan and Bey, 2004). The study by Yu et al. (2013) found that the long-range transport contributes by 6-16% and 22-40% to aerosol optional depth and direct radiative forcing in 4 regions including North America, Europe, East Asia and South Asia. Recent studies have reported an increasing trend in the hemispheric transport of air pollution. In particular, the air pollution exported from Asia to North America has increased significantly in recent years (Jaffe et al., 2003; Parrish et al., 2009; Parrish et al., 2004; Verstraeten et al., 2015; Zhang et al., 2008) due to the rapid growth of Asian emission (Richter et al., 2005; Lu et al., 2010; Zhang et al., 2007; van der A et al., 2006; van der A et al., 2008).

Compared to the impact on air pollution, the impact of hemispheric transport on deposition hasn't been fully studied. Arndt and Carmichael (1995) developed a source-receptor (S-R) relationship for S deposition among the Asia regions in early 1900s. Zhang et al. (2012) found foreign anthropogenic emission contributes to 6% and 8% of the $NO_y$ and $NH_x$ deposition in contiguous United States, respectively. A systematic study by Sanderson et al. (2008) shed light on the impact of long-range transport on deposition of oxidized nitrogen at global scale. That study uses the model ensemble results from phase I of Task Force Hemispheric Transport of Air Pollution (HTAP I) to calculate the S-R relationship for $NO_y$ deposition in 2001 among 4 regions: Europe, North America, South Asia and East Asia. Results showed that about 12-24% of the $NO_x$ emission is deposited out of source regions. About 3-10% of the emission is deposited on the other 3 regions and affects their deposition by about 1-3%. However, these studies focused on emission intense regions, where the foreign disturbance is relatively small compared to huge own region emission. The foreign impact on low emission regions was not evaluated in the same detail. Furthermore, both the magnitude and spatial distribution of S and N emission and deposition have been changed considerably during the last 10 years (2001-2010)





(Tan et al., 2018). It is necessary to update the S-R relationship for more recent years with regards to these changes.

To explore these questions, this study assesses the impact of hemispheric transport of S, $NO_x$ and $NH_3$ emissions on S and N deposition, with multi-model ensemble results from $2^{nd}$ phase of HTAP (HTAP II). Additional to the 4 regions: North America (NA), Europe (EU), South Asia (SA), East Asia (EA) used in Sanderson's (2008) study for HTAP I, we include 2 regions: Middle East (ME) and Russia, Belarussia, Ukraine (RU) in this study. These two regions have low S and N emissions relative to their areal extent, but are located close to high emission regions such as EU, SA and EA. We calculate the amount of deposition brought by hemispheric transport by comparing model results for base case and for 20% emission perturbation cases. The experimental design is described in Section 2. Section 3 is the result part and has 3 subsections. We explore the following questions:

1) Which fraction (percentage) of the S or N emissions is transported and deposited outside of this region? Which fraction is finally deposited on the other 5 receptor regions, other regions and oceans? What is the seasonality of the exported fraction?

2) As receptor regions, what is the amount of deposition brought by hemispheric transport? How much will it affect the local deposition? Is there any seasonality for this impact?

3) For each region, what are the contributions of hemispheric transport from foreign regions and of control of own region emission on deposition? In line with the analysis for other pollutants, to this purpose we evaluate the so-called response to extra-regional emission reduction (RERER) metric. We also discuss the own region and foreign impact specifically on the coastal regions. The inter-model variations are also illustrated in this section.

Section 4 is a summary of the findings in this study and some suggestions for future study.

**2 Methodology**

**2.1 HTAP II and experiment set-up**

The HTAP was created in 2004 under the Convention on Long-range Transboundary Air Pollution (CLRTAP).  The project involves efforts from international scientists aiming at understanding the hemispheric transport of air pollutants and its impact on regional and global air quality, public health and near-term climate change. Until now, two phases of HTAP experiments have been conducted successfully. The HTAP I involved more than 20 models from



international modelling groups, with 2001 as the base year for modeling studies. A
comprehensive report of the major findings of HTAP was released in 2010 and could be
downloaded from http://www.htap.org/. The HTPA II was launched in 2012, with 2010 as the
base simulation year. HTAP II required all models to use the same prescribed anthropogenic
emission inventory and boundary conditions instead of using the best estimates of emission by
each model group as HTAP I, which facilitated an inter-model comparison between models. In
addition, HTAP II had a refined definition for the boundaries of regions, which enabled an
update in the S-R relationships for air pollutants and deposition among regions.
This study uses the ensemble of 11 global models from HTAP II (including CAM-Chem,
CHASER_re1, CHASER_t106, EMEP_rv48, GEMMACH, GEOS5, GEOSCHEMAJOINT,
OsloCTM3v.2, GOCARTv5, SPRINTARS and C-IFS_v2). A detailed description of the
experiment set-up could be found in Galmarini et al. (2017). The S deposition includes $SO_2$
deposition and $SO_4^{2-}$ deposition. N deposition is categorized to oxidized nitrogen ($NO_y$) and
reduced nitrogen ($NH_x$) deposition. $NO_y$ deposition is a sum of all oxidized N except $N_2O$,
including $NO_2$, $HNO_3$, $NO_3^-$, PAN and other organic nitrates than PAN (Orgn). $NH_x$ deposition
includes $NH_3$ deposition and $NH_4^+$ deposition. To form the multi-model ensemble, we re-grid all
models to a uniformed horizontal resolution of $0.1° \times 0.1°$. We use the multi-model mean value
(MMM) of all models to present the ensemble results, a procedure which has previously been
proven to have a better agreement with observations than single model results (Dentener et al.,
2006; Tan et al., 2018). The mean values of the compositions of S or N deposition are calculated
separately and then are combined to compute the total S or N deposition. More details can be
found in Tan et al. (2018). The inter-model variations are discussed in section 3.3.
**2.2 Simulation scenarios**
The base simulation uses anthropogenic emissions in 2010 (Janssens-Maenhout et al.,
2015), which is called "base case" in this study. The MMM performance on wet deposition has
been evaluated with observations from National Atmospheric Deposition Program (NADP)
(http://nadp.sws.uiuc.edu/) for NA, European Monitoring and Evaluation Programme (EMEP)
CCC reports (http://www.nilu.no/projects/ccc/reports.html) for EU and Acid Deposition
Monitoring Network in East Asia (EANET) (http://www.eanet.asia/) for EA in Tan et al. (2018).
In terms of wet deposition, MMM results are evaluated with site observation of deposition. $SO_4^{2-}$





wet deposition is generally well simulated with 76% of the stations of all networks predicted
within ±50% of observation. Negative model biases (-20%) are found at some East Asian
stations. Modeled $NO_3^-$ wet deposition is within ±50% of observation for 83% of the stations of
all networks. The European and Southeast Asian stations with high observed $NO_3^-$ wet deposition
are found to be somewhat underestimated. 81% of modeled $NH_4^+$ wet deposition at stations of all
networks are within ±50% of observation. A general underestimation is found in modelled $NH_4^+$
wet deposition, especially at East Asian stations. In terms of dry deposition, due to the lack of
directly measured dry deposition, we compare the modeled dry deposition with inferential data
from the Clean Air Status and Trends Network (CASTNET) over United States. The CASTNET
inferential data is calculated with observed aerosol concentration and modelled dry deposition
velocity, therefore it has high uncertainty in data quality. Results show that the modelled dry
deposition is generally a factor of 1-2 higher than the CASTNET inferential data. This is a
common feature of many global and regional models (WMO, 2017) and subject to further
research.

In addition to the base case simulations, emission perturbation experiments are conducted

for certain regions. The boundaries of 17 regions in HTAP II are defined in Fig. 1. In the
perturbation experiments, the anthropogenic emissions (including $NO_X$, $SO_2$, $NH_3$, VOC, CO
and PM) of specific regions (i.e. NA) are reduced by 20% from the amounts in the base case
simulation, while the other regions keep the same emissions. This study uses the perturbation
experiments of 6 regions (with color in Fig.1) with high priority: NA, EU, SA, EA, ME and RU.
In addition, a global perturbation experiment referred as "GLO" is conducted with 20% decrease
in global anthropogenic emissions. We estimate the impact of hemispheric transport on
deposition by comparing the model results under perturbation experiments with those under base
case simulation. In order to validate the quality of model outputs, we check the mass balance
between emission and deposition at global scale. The mass balance for base case simulation has
been checked in Tan et al. (2018), therefore we show the mass balance for perturbation
experiments in this study. We compare the global total amounts of changes of deposition (Δ
Depo) with changes of emissions (Δ Emis) for all perturbation cases (Table S1). According to
our results, the amounts of Δ Depo are almost identical to Δ Emis for all perturbation cases. The
Δ Depo of $NH_x$ deposition under EA perturbation case is not available due to that no model
meets the mass balance requirements mentioned above.



## 3 Results

### 3.1 Export of S and N emissions from source regions

This section studies the export of S and N emissions and oxidation products from source regions. Table 1 shows the S-R relationship of S and N deposition among the 6 regions. The numbers are the sensitivity ($SEN_{r \to s}$) of deposition in the receptor/source regions to emission changes in the source regions (Sanderson et al., 2008). The metric is calculated as $\Delta$ Depo in the receptor/source regions divided by $\Delta$ Emis in the source regions following equation (1).

$$SEN_{r \to s} = \frac{\Delta \, Depo \, (r/s)}{\Delta \, Emis \, (s)} \times 100\% \qquad (1)$$

where s is the source region and r is the receptor region. $\Delta$ Depo (r/s) is the deposition change in the receptor/source regions, $\Delta$ Emis (S) is the emission change in the source regions. This value indicates the fraction of emission from source regions that is deposited locally or exported to foreign regions.

The numbers in Table1 are for coastal and non-coastal regions together and the numbers in the parenthesis are specifically for coastal regions (defined in Fig. 1). "Others" means the other regions in the world than the 6 regions (white color in Fig.1). The NA region has 69% of its S emission deposited within itself, including 9% deposited on its coastal region. The remaining 31% is exported to the other regions, mostly to the "Others" and less than 1% is deposited on the other 5 regions (EU, SA, EA, ME and RU). A relatively large fraction (14 %) of European S emissions are exported to RU region. Other major pathways of export of sulphur emissions/reaction products are from SA to EA (9%), from EA to RU (5%) and from RU to EU (7%) and EA (5%). ME has considerable percentages of S emission exported to its nearby regions such as SA (8%), EA (5%) and RU (5%). The S-R relationship of $NO_y$ deposition is similar to that of S deposition, except that EU and ME have 66% and 54% of $NO_x$ emissions deposited within the source region, which are 6% and 12% higher than those of S emissions, likely due to the large average emission altitude of S emissions and somewhat longer lifetimes compared to NOx emissions. In terms of $NH_x$ deposition, about 20% more $NH_3$ emission is deposited within the source region due to its short lifetime in the atmosphere.

The seasonal variation of the export of S and N emissions by source regions is shown in Fig. S1. In terms of S emission, there is 5-10% of seasonal variation around the annual average



of 25% in the export fractions for all regions except SA. SA exports almost half of its annual S
emission in spring, which is twice the numbers in summer (20%) and fall (25%), related to the
specific dry period and monsoon circulation. The seasonal export fractions of $NO_x$ and $NH_3$
emission are similar to that of S emission, but generally lower in all seasons. Generally, the
source regions export highest percentage of their emissions in winter and spring and lowest in
summer. More proficient oxidation chemistry in summer resulting in more soluble component,
and local weather systems, especially the episodic of precipitation has large influence on this
seasonality. For most continental regions, the wet deposition accounts for 50-70% of total
deposition (Tan et al., 2018; Vet et al., 2014; Dentener et al., 2006). Therefore, local
precipitation washes out large proportion of the air pollutants in the atmosphere during the
rainfall seasons (i.e. summer). In addition, the strong westerly winds in winter and spring favor
the hemispheric transport, while the rapid vertical convection in summer slows down the zonal
transport of air flows and accelerates the removal process.
We compare our results with previous studies. Bey et al. (2001) estimated that 70% of
emitted $NO_x$ from Asia is lost within its boundary by deposition of $HNO_3$ in spring. The
estimation is our study is 70% for EA and 61% for SA, close to Bey's result. Li et al. (2004)
reported that about 20% of anthropogenic $NO_x$ emitted by NA is deposited out of its boundary
(about 1000 km offshore). Stohl et al. (2002) calculated that 9-22% of surface $NO_y$ emissions is
exported out of the boundary layer of NA. Our result is about 30%, higher than Li's and Stohl's
results. HTAP I study by Sanderson et al. (2008) developed a SR relationship for $NO_y$ deposition
among NA, EU, SA and EA. About 12-24% of the emitted $NO_x$ is deposited out of source
regions and the corresponding percentage in this study is 26-34%. It should be noted that the
estimations of different studies are influenced by several factors and the results are not fully
comparable. (1) Definition of boundaries of source and receptor regions. For instance, Li et al.
(2004) defined the boundary of NA by a squared domain: 65-130°W, 25-55°N, while we use the
continental boundaries defined by HTAP II. There are also changes in the definition of
boundaries between HTAP II and HTAP I. For instance, HTAP I includes Mexico and Central
America in NA, but they are defined as a separate region in HTAP II (region 12 in Fig. 1). The
boundary of EU is also changed in HTAP II. (2) HTAP I used the perturbation simulation that
only reduce the $NO_x$ emission, but HTAP II simulations also reduce other anthropogenic





emissions, including $SO_2$ and PM. The joint control of multiple emissions may cause more
reduction in $NO_y$ deposition and it is hard to estimate this effect in this study.

**3.2 Impact of hemispheric transport on deposition**

This section investigates the impact of hemispheric transport on deposition in the receptor
regions. Fig. 2 is annual response of S deposition to 20% emission reduction in source regions,
calculated as (Δ Depo under perturbation case) / (deposition in base case) ×100%. The negative
values mean that the corresponding deposition decreases with reduction in emission. Table S2
summarizes the regional median deposition fluxes under base case and under emission
perturbation in source regions. Fig. 2(a) shows the global response of deposition to 20%
emission reduction in NA. The largest changes happen in the source region NA, with a 5-20%
decrease in S deposition in the non-coastal region and 15-20% decrease at the east coast. The
impact on the North Atlantic Ocean deposition declines gradually from near coastal region (12-
15%) to open ocean (5-12%) and Eurasia (<1%). Fig. 2(b) shows the global response of
deposition to 20% emission reduction in EU. Although the impact on continental non-coastal
regions is high (12-20%), the impact on the coastal regions is within 5%, much lower than NA's
impact on its east coast (15-20%). The deposition in North Africa, central Asia and western RU
is affected by 2-5%. 20% emission reduction in SA (Fig. 2(c)) shows large influence over its
south-west coast and the Arabian Sea (5-12%). The SA's outflow affects the deposition in
southeastern ME and eastern Sub Saharan Africa by 1-5% and western EA and Southeast Asia
(Bangladesh) by 2-10%. Fig. 2(d) shows the impact of 20% emission reduction on deposition in
EA. On one hand, the impact is strong on the east coast of China (12-15%) and decreases
gradually over the North Pacific Ocean (5-15%). Although the majority of S emission is
deposited on the Pacific Ocean, the influence on western NA can still reach 5-8%. On the other
hand, the impact on Southeast Asia and South Asia is much lower (2-5% and <1%), due to the
block of air flows by the Himalaya Mountains (Fig. S4). 20% emission decrease in ME mainly
affects the S deposition in Africa by 2-10% and western EA by 2-5%. Fig. 2(f) shows the S
deposition change with 20% emission reduction in RU. The regions of impact are mainly at high
latitudes, including northern EU (2-5%) and western Arctic Circle (1-5%).
The impact of $NO_x$ emission reduction on $NO_y$ deposition from each source region is
shown in Fig 3. The overall impact is qualitatively similar to that of S emission with some



differences. For some regions, the impact of hemispheric transport on $NO_y$ deposition is lower
than that on S deposition. For instance, SA's impact on eastern Africa is about 2-5% on S
deposition, but is <1% on $NO_y$ deposition. ME's impact on the east coast of Africa and Gulf of
Guinea is about 2-5% on S deposition, but is <1% on $NO_y$ deposition. These smaller sensitivities
reflect differences in lifetimes, and the lower formation of aerosol nitrate under warm conditions
in tropical regions. Under the NA perturbation case (Fig. 3(a)), a 2-12% change of $NO_y$
deposition is found on the west coast of California, due to high $NO_x$ emission in California from
mobile source, which is not seen in S deposition. The impact of emission reduction in EU and
EA on their coastal regions is generally 2-5% higher for $NO_y$ deposition than S deposition (Fig.
3(b) and (d)). The impact on $NH_x$ deposition is similar to $NO_y$ deposition (Fig. S2). It should be
noted that this is the result from 20% emission reduction in the source regions, therefore the
actual impact (100% emission reduction) could be 5 times higher when assuming a linear
relationship between 20 and 100% emission reduction on deposition.

We quantify the amount of deposition carried by hemispheric transport and study its

seasonality. Fig. 4 shows the monthly changes of S deposition for 20% emission reductions in
source regions. The values are meridional sum with a west-east resolution of 0.1 degree, and
display well the locations of the source regions. The negative values indicate the amounts of
pollutants transported from source regions to receptor regions. According to Fig 4(a), NA has
about $(1-10) \times 10^4$ kg(S) month$^{-1}$ per 0.1° longitude of its S emission transported and deposited
over the North Atlantic Ocean (15-75°W) throughout the whole year. We also find about (1-3)
$\times 10^4$ kg(S) month$^{-1}$ per 0.1° longitude decrease of S deposition at about 90°E and 120°E in
spring and fall, which gives evidence to NA's influence on Eurasia via transatlantic flow,
although this amount accounts for less than 1% of local S deposition (white in Eurasia in
Fig.2(a)). Fig. 4(b) shows that about $(1-3) \times 10^4$ kg(S) month$^{-1}$ per 0.1° longitude of EU's
emission is transported and deposited at 30-60°E in RU throughout the whole year and at 100-
120°E in EA in spring and fall. According to Fig. 4(c), SA exports its S emission to 30-60°E in
ME and eastern Africa in early spring and to 90°E-180° in EA and North Pacific Ocean from late
spring until fall. In particular, the influence on EA can reach $(5-10) \times 10^4$ kg(S) month$^{-1}$ per 0.1°
longitude in mid-spring. According to Fig. 4(d), EA's S emission is widely transported and
deposited over the North Pacific Ocean throughout the whole year. The Asian outflow arrives at
the west coast of NA (~130°W) in all seasons except summer, but only reaches far in western





NA (~90°W) in spring and brings about $1 \times 10^4$ kg(S) month$^{-1}$ per 0.1° longitude of S deposition.
The monthly changes of $NO_y$ deposition with perturbation experiments are shown in Fig. 5.
Compared to S deposition, the change in $NO_y$ deposition by hemispheric transport is generally
smaller. For instance, the NA's impact on Eurasia is $(1-3) \times 10^4$ kg(S) month$^{-1}$ per 0.1° longitude
for S deposition, but is less than $0.5 \times 10^4$ kg(N) month$^{-1}$ per 0.1° longitude for $NO_y$ deposition.
The SA's impact on EA (90-120°E) can reach $(5-10) \times 10^4$ kg(S) month$^{-1}$ per 0.1° longitude for S
deposition, but the amount is 4 times lower for $NO_y$ deposition. This result is in accordance with
the S-R results in section 3.1 that more S emission is transported out of the source regions than N
emission, due to higher emission altitudes and longer chemical lifetimes. Patterns similar to $NO_y$
are also found in the monthly changes of $NH_x$ deposition (Fig. S3).
The deposition change via transport between neighboring regions is found throughout the
whole year and is slightly stronger in winter, such as between EU and RU (~30°E) (Fig. 4(b) and
(f)) and from EA to the North Pacific Ocean (~130°E) (Fig. 4(d)). This is consistent with the
seasonality we found for the export of emission by source regions in section 3.1. In addition,
most source regions reduce more S and $NO_x$ emissions in winter than the other seasons (Table
S3), thus more emissions are exported abroad in winter. On the contrary, most of the deposition
change by transport over long distance occurs in spring and fall, especially for the hemispheric
transport from NA to EU, from EU to EA and from EA to NA. This agrees with the seasonality
of the transpacific, transatlantic and trans-Eurasia flows of air pollutants (Holzer et al., 2005;Liu
et al., 2005;Liang et al., 2004;Brown-Steiner and Hess, 2011;Li et al., 2014;Auvray and Bey,
2005;Wild et al., 2004;Liu et al., 2003). The long distance transport of emissions is low in winter,
due to that the formation of secondary species like PAN is suppressed due to slow oxidation
(Berntsen et al., 1999;Deolal et al., 2013;Moxim et al., 1996), which plays an important role as a
reservoir for $NO_x$ in the long-range transport of air pollution (Lin et al., 2010;Hudman et al.,

2004).

**3.3 Own region and foreign contributions on deposition**

This section compares the contributions of hemispheric transport and own region emission
control on deposition. The second metric is the response to extra-regional emission reduction
(RERER) as shown in Table 2. It is calculated by dividing the Δ Depo due to foreign emission





reduction by Δ Depo due to global (foreign + own region) emission control following equation

(2).

$$RERER_i = \frac{\Delta Depo_i \,(foreign)}{\Delta Depo_i \,(global)} \qquad (2)$$
where i is the region of focus. $\Delta Depo_i$ (foreign) is the Δ Depo in region i due to 20% foreign
emission reduction. It is calculated by subtracting the Δ Depo due to 20% own region emission
change from Δ Depo due to 20% global emission change. $\Delta Depo_i$ (global) is the Δ Depo in
region i due to 20% global emission reduction. This metric compares the contributions from
foreign emission reduction with own region emission control on local deposition. A low RERER
value (close to 0) indicates a predominance impact of own region emission on local deposition,
while high RERER value (close to 1) means strong impact of hemispheric transport on local
deposition.

The total column includes both non-coastal and coastal regions. As we expected, NA

(0.07-0.17), SA (0.04-0.18) and EA (0.16) regions have relatively low RERER values, due to
large emissions and deposition in those regions compared to the foreign contributions. EU (0.12-
0.36) and ME (0.32-0.42) have relatively higher RERER values. RU is the only region with
RERER (0.55-0.61) higher than 0.5, which means its deposition is almost equally sensitive to
foreign impact and own region control. The RERER values of S deposition and $NO_y$ deposition
are of similar magnitudes, while the RERER of $NH_x$ deposition is 0.1 lower, due to the lack of
contribution from EA.

For non-coastal regions, the own region impact includes control of both its coastal and

non-coastal regions. The foreign impact comes from emission reduction of foreign coastal and
non-coastal regions. The RERER values of coastal regions are generally 0.1-0.3 higher than
those of non-coastal regions. In particular, the values for RU's coast are all higher than 0.84.
Even regions with low total RERER such as NA and SA have high RERER on coastal regions.
For instance, the RERER of NA reached 0.3-0.4 for its coastal region, more than double of the
RERER on its non-coastal regions (0.05-0.12). Coastal regions receive high proportion of
deposition from foreign transport. According to table 1, EA exports 5% of its S and N emission
to RU, almost half of which is deposited on RU's coastal regions. RU exports 7-12% of S and N
emission to EA, of which 30% is deposited on EU's coastal regions. The impact of hemispheric



transport is identical or even larger than the effect of own region emission control for some
coastal or near coastal regions. According to Fig. 2, 20% emission reduction in EA can reduce 2-
5% of S deposition in the west coast of NA. This effect is even larger than 20% emission
reduction in NA (<1%). Similarly, 20% emission reduction in NA can change 2-5% of S
deposition in west coast of EU, which is almost identical to the effect of 20% emission control in
EU. On one hand, the emissions in western NA and western EU are relatively low, thus the effect
of own region control is not significant. On the other hand, these coastal regions are in the
downwind location of eastern EA and eastern NA, which are the main source regions of S and N
emissions. Coastal regions serve as transit places for air-sea exchange with vulnerable ecosystem
(Jickells, 2006;Jickells et al., 2017). The over-richness of deposition in coastal water and
ecosystem can evoke a number of environmental issues, of which some are specifically for
coastal regions such as threats to coastal benthic and planktonic system and sustainability of
fishery (Paerl, 2002;Doney et al., 2007).
Figure 6 compares the foreign and own region contributions on own region deposition.
Other (OTH, pattern fill in the figure) is calculated as $\Delta \text{Depo}_{(\text{GLO})} - \sum \Delta \text{Depo}_{(\text{case})}$ (case = 6,
including NA, EU, SA, EA, ME and RU). It indicates the deposition change in the receptor
region due to other reasons than the sum of separate emission reduction in the 6 regions. For
regions with low RERER values (NA, SA and EA), the own region emission dominates the
deposition by more than 80%. For these regions, determined by vicinity and transport patterns,
the foreign impact is somewhat dominated by certain source regions, such as from EA to NA (2-
4% out of 4-5%), from ME to SA (5-6% out of 7-11%) and from SA to EA (3-4% out of 4-7%).
For EU and ME, there is about 20% contribution from "OTH". It could come from the emission
reduction in rest of world, especially nearby regions such as Central Asia and North Africa. It
could also come from the joint effects of emission control by multiple source regions. However,
the model simulations do not allow to separate these two contributions in this study. Beside this,
RU contributes 4-5% to EU's deposition and EU contributes 5% to ME's deposition. For high
RERER regions, RU has a different pattern than the other regions. The contributions of
hemispheric transport from other 5 regions (23-45%) are almost equivalent to its own region
emission control (39-45%). There are significant contributions from EA (20-24%) and EU (13-
15%), which is reasonable since RU emits low S and N emissions, but is located close to these
two major source regions.



Fig. 7 shows the inter-model variation on simulating the changes of deposition of $SO_x$,

$NO_y$, and $NH_x$ under emission perturbation cases, separate for wet and dry deposition. The

values are global integrated changes in component deposition for perturbation experiments from

MMM results with error bars showing the maximum and minimum values of models. The figure

only shows main compositions of S and N deposition, which together account for more than 95%

of total deposition. In terms of S deposition (Fig. 7(a)), the differences between the maximum

and minimum values of models for different perturbation cases range from $\pm$(0.03-0.13) Tg(S)

($\pm$29-83% of MMM), $\pm$(0.00-0.11) Tg(S) ($\pm$39-78%), $\pm$(0.00-0.01) Tg(S) ($\pm$21-57%) and $\pm$(0.01-

0.09) Tg(S) ($\pm$11-51%) for $SO_2$ dry and wet deposition and $SO_4^{2-}$ dry and wet deposition,

respectively. High uncertainty is found in EA perturbation case, where the model divergence are

mainly found on $SO_2$ wet and dry deposition and $SO_4^{2-}$ wet deposition. In terms of $NO_y$

deposition (Fig. 5(b)), the differences among cases are about $\pm$(0.00-0.05) Tg(N) ($\pm$16-39%),

$\pm$(0.02-0.28) Tg(N) ($\pm$23-70%) and $\pm$(0.02-0.47) Tg(N) ($\pm$15-45%) for $NO_2$ dry deposition and

$NO_3^-$ dry and wet deposition, respectively. The EA case also has the largest inter-model

variation, with high uncertainty in simulating both the $NO_3^-$ wet and dry deposition. In terms of

$NH_x$ deposition (Fig. 5(c)), the differences are about $\pm$(0.00-0.06) Tg(N) ($\pm$2-27%), $\pm$(0.00-0.08)

412       Tg(N) ($\pm$10-55%) and $\pm$(0.01-0.06) Tg(N) ($\pm$5%) for $NH_3$ dry deposition and $NH_4^+$ dry and wet

deposition, respectively. Both EA and SA have relatively high uncertainties on $NH_4^+$ dry

deposition. Overall, the inter-model variation is considerably high for the deposition change

under EA emission perturbation. On one hand, the EA perturbation case assumes the largest

amount of emission reductions among all cases (Table S3). On the other hand, model evaluation

(Tan et al., 2018) also has reported high model bias in simulating the deposition in this region,

and suggest an incomplete knowledge from the combined picture provided by observations and

models.

**4 Conclusion**

This study assesses the impact of hemispheric transport on S and N deposition for 6 regions:

North America, Europe, South Asia, East Asia, Russia and Middle East, by using multi-model

ensemble results from 11 models of HTAP II project, with simulations under base case and 20%

emission reduction scenarios.



We investigate the export of S and N emissions by source regions. Results show that about 27-58%, 26-46% and 12-23% of the emitted S, $NO_x$ and $NH_3$ emissions are deposited outside of the source regions. The most significant exports of emissions are from Europe to Russia (10-14%), from South Asia to East Asia (4-9%), from East Asia to Russia (5%) and from Russia to Europe (7-12%) and East Asia (4-5%). Most regions export 5-10% more emission to abroad in winter than summer, which is highly influenced by chemistry, precipitation amount and frequency, atmospheric mixing and transport patterns.

We explore the impact of hemispheric transport on deposition in receptor regions. Overall, 20% emission reduction in source regions could affect 1-10% of deposition on foreign continental regions and 1-15% on foreign coastal regions and open ocean, especially from North America to the North Atlantic Ocean (5-15%), from South Asia to western East Asia (2-10%) and from East Asia to North Pacific Ocean (5-15%) and western North America (5-8%). The amounts of deposition brought by hemispheric transport range from $10^4$-$10^5$ kg(S or N) month$^{-1}$ per 0.1° longitude (meridional sum). The impact on deposition via transport between neighbouring regions (i.e. Europe and Russia) is generally found throughout the whole year and slightly stronger in winter, while that via transport over long distance (i.e. from East Asia to North America) mainly takes place in spring and fall.

We compare the own region and foreign impact on deposition. The deposition in North America, South Asia and East Asia is dominated (~80%) by their own region emission, while Europe, Middle East and Russia receive 40-60% of impact from hemispheric transport. In particular, Russian deposition is even equally sensitive to foreign inputs and own region emission, with high contributions from two neighbouring source regions: East Asia (~20%) and Europe (~15%). Coastal regions receive upmost half of the hemispheric transport from foreign regions. Deposition in coastal regions or near-coastal open ocean is found twice more sensitive to long-range transport than non-coastal regions. For some coastal regions such as west coast of North America and west coast of Europe, the impact of hemispheric transport is identical or even larger than that of own region emission control.

This study highlights the impact of hemispheric transport on deposition in coastal regions and open ocean, which hasn't been fully studied in the literature. We therefore suggest further research on this impact on the mitigation of coastal and oceanic ecosystem, with regards to the increasing concentration of air pollutants in hemispheric outflow. We also find significant impact



of hemispheric transport on deposition in relatively low emission regions such as Russia. The
impact on their ecosystem and human health requires further research. Meanwhile, there is still a
portion of foreign impact that hasn't been attributed in this study (aggregated as other regions
"OTH" in Fig. 6). Some regions are not included in perturbation experiments, but are found with
large impact by/to other regions. For instance, at least 4 regions (North America, Europe, South
Asia and Middle East) have shown considerable impact (2-10%) on the S and N deposition in
North Africa. The impact from/by Southeast Asia is also unknown, which is regarded as a big
contributor of global S and N emissions in Asia. We suggest the future HTAP simulations to
include these regions in the perturbation experiments.

*Acknowledgements*. We thank all participating modelling groups in HTAP II for providing the
simulation data. The National Center for Atmospheric Research (NCAR) is funded by the
National Science Foundation. The CESM project is supported by the National Science
Foundation and the Office of Science (BER) of the U.S. Department of Energy. Computing
resources were provided by the Climate Simulation Laboratory at NCAR's Computational and
Information Systems Laboratory (CISL), sponsored by the National Science Foundation and
other agencies. Supercomputer system of the National Institute for Environmental Studies, Japan.
The Environment Research and Technology Development Fund (S-12-3) of the Ministry of the
Environment, Japan. JSPS KAKENHI grants 15H01728.



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



Caption:
Fig. 1. Administration boundaries of regions and coastal areas. 6 Regions with perturbation
experiments: 3-North America (NA), 4-Europe (EU), 5-South Asia (SA), 6-East Asia (EA), 11-
Middle East (MD) and 14-Russia, Belarussia, Ukraine (RU). Other regions: 1-Global, 2-Ocean
(including Arctic), 7-Southeast Asia, 8-Australia, 9-North Africa, 10- Sub Saharan Africa, 12-
Mexico, Central America, Caribbean, Guyanas, Venezuela, Columbia (Central America), 13-
South America, 15-Central Asia and 17-Antarctic.
Fig. 2 The response of S deposition to 20% emission reduction in source regions. The values are
the percentage changes (%) in deposition calculated as (changes in deposition with 20%
emission reduction) / (base case deposition) ×100%. The unit is %.
Fig.3 Same as Fig.2 but for $NO_y$ deposition. The unit is %.
Fig. 4 The monthly changes of S deposition with 20% emission reduction in source regions. The
values are meridional total values versus time with a west-east resolution of 0.1 degree. The unit
is $\times10^4$ kg(S) month$^{-1}$ per 0.1° longitude. The negative values indicate decline in deposition with
reduction in emission.
Fig. 5 Same as Fig.4 but for $NO_y$ deposition. The unit is $\times10^4$ kg(N) month$^{-1}$ per 0.1° longitude.
Fig. 6 Own region and foreign contributions on own region deposition. The values are calculated
by changes with 20% emission reduction. Other (OTH, pattern fill) is the contribution by other
reasons than emission reduction in the 6 regions (see text for details).
Fig. 7 Inter-model variations in deposition changes (unit: Tg(S or N) yr$^{-1}$) under emission
perturbation experiments. The values are MMM with error bars showing the max and min values
among all models.



**Fig. 1**

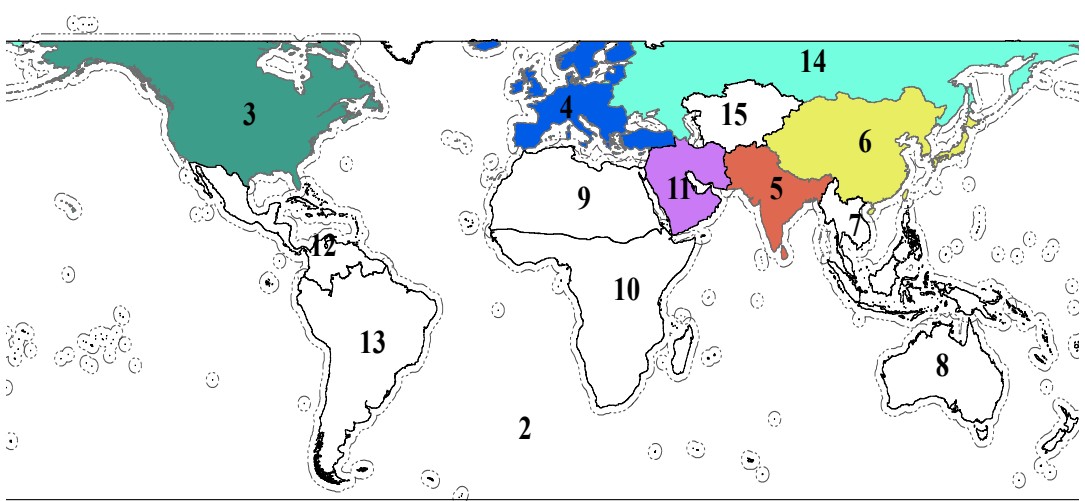

Fig. 1. Boundaries of regions and coastal areas (dashed). 6 Regions with perturbation
experiments: 3-North America (NA), 4-Europe (EU), 5-South Asia (SA), 6-East Asia (EA), 11-
Middle East (MD) and 14-Russia, Belarussia, Ukraine (RU). Other regions: 1-Global, 2-Ocean
(including Arctic), 7-Southeast Asia, 8-Australia, 9-North Africa, 10- Sub Saharan Africa, 12-
Mexico, Central America, Caribbean, Guyanas, Venezuela, Columbia (Central America), 13-
South America, 15-Central Asia and 17-Antarctic.

**Fig. 2**

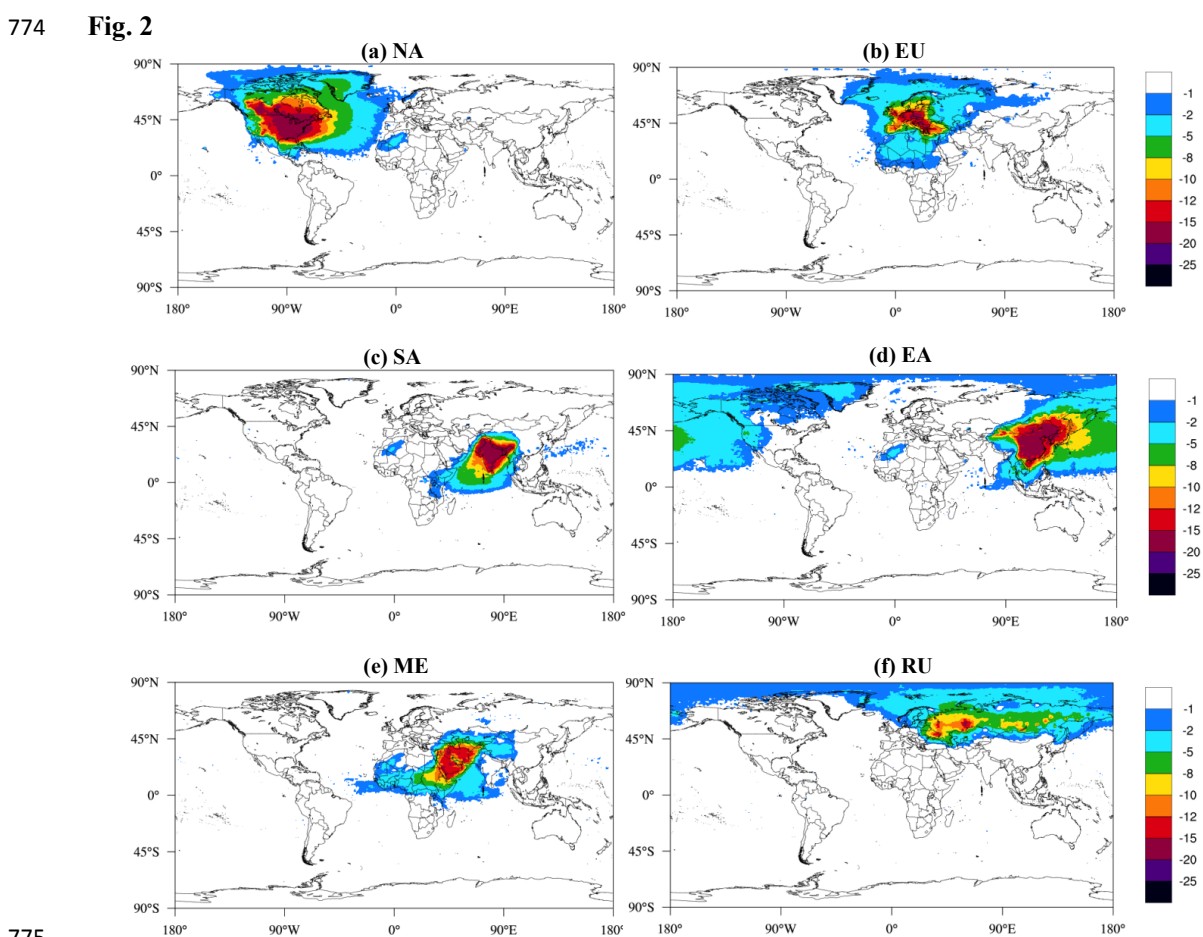

Fig. 2 The response of S deposition to 20% emission reduction in source regions. The values are
the percentage changes (%) in deposition calculated as (changes in deposition with 20%
emission reduction) / (base case deposition) ×100%. The unit is %.



**Fig. 3**

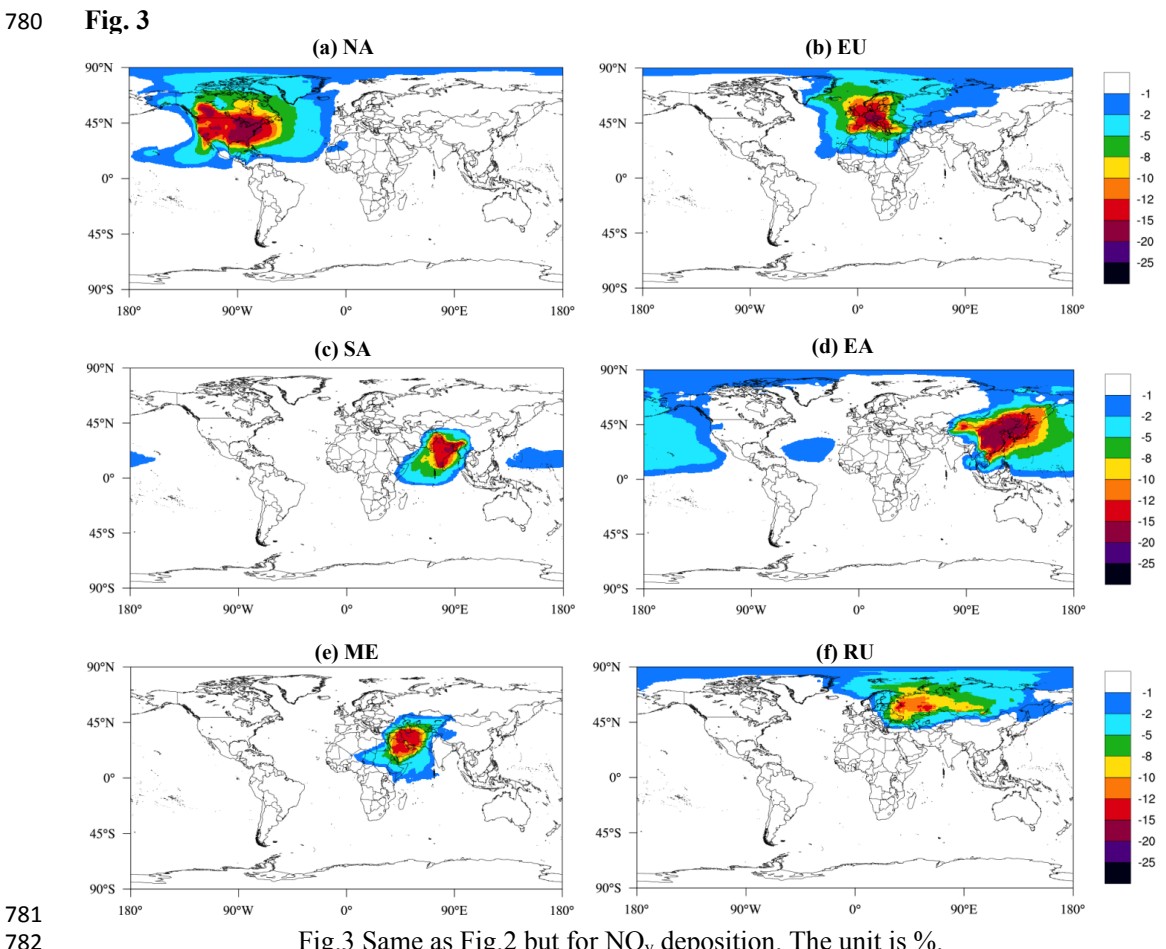

Fig.3 Same as Fig.2 but for NO$_y$ deposition. The unit is %.



**Fig. 4**
Fig. 4 The monthly changes of S deposition with 20% emission reduction in source regions. The
x-axis values are meridional total values versus time (y-axis) with a west-east resolution of 0.1
degree. The unit is $\times 10^{4}$ kg(S) month$^{-1}$ per 0.1° longitude. Negative values indicate decline in
deposition with reduction in emission.



**Fig. 5**
Fig. 5 Same as Fig.4 but for $NO_y$ deposition. The unit is $\times 10^4$ kg(N) month$^{-1}$ per 0.1° longitude.




**Fig. 6**
Fig. 6 Own region and foreign contributions on own region deposition. The values are calculated
by changes with 20% emission reduction. Other (OTH, pattern fill) is the contribution by other
reasons than emission reduction in the 6 regions (see text for details).





Fig. 7



Fig. 7 Inter-model variations in wet and dry deposition changes (unit: Tg(S or N) yr$^{-1}$) under emission perturbation experiments. The values are global integrated changes in components of S and N deposition for perturbation experiments from MMM results with error bars showing the max and min values among all models.





Tables
Table 1. Source-receptor relationship of S/$NO_y$/$NH_x$ deposition (%) for regions (including
continental coastal and non-coastal regions). The values in the parentheses are for coastal regions
as a subset of the total.

| | Receptor Regions | Source Regions | | | | | |
|---|---|---|---|---|---|---|---|
| | | NA | EU | SA | EA | ME | RU |
| S Deposition | NA | 68.9 (8.9) | 0.2 (0.1) | 0.2 (0.0) | 1.5 (0.6) | 0.3 (0.1) | 1.2 (0.6) |
| | EU | 1.1 (0.6) | 60.4 (14.4) | 0.0 (0.0) | 0.2 (0.1) | 2.1 (0.2) | 6.9 (2.5) |
| | SA | 0.5 (0.1) | 1.2 (0.3) | 66.4 (10.0) | 0.9 (0.4) | 7.9 (1.6) | 0.3 (0.1) |
| | EA | 0.6 (0.2) | 1.8 (0.4) | 8.8 (1.3) | 73.4 (11.5) | 4.6 (0.8) | 5.2 (1.4) |
| | ME | 0.0 (0.0) | 2.6 (0.6) | 0.6 (0.3) | 0.0 (0.0) | 42.4 (8.2) | 0.8 (0.2) |
| | RU | 0.4 (0.1) | 13.6 (2.2) | 0.1 (0.1) | 5.1 (2.2) | 5.0 (1.1) | 62.2 (4.4) |
| | Others | 28.5 | 20.1 | 23.8 | 19.1 | 37.7 | 23.4 |
| $NO_y$ Deposition | NA | 71.5 (7.8) | 0.8 (0.2) | 0.5 (0.1) | 1.0 (0.3) | 0.5 (0.1) | 1.0 (0.3) |
| | EU | 1.3 (0.6) | 66.2 (17.5) | 0.2 (0.1) | 0.3 (0.1) | 3.5 (0.9) | 9.8 (2.9) |
| | SA | 0.2 (0.0) | 0.2 (0.0) | 66.2 (8.6) | 0.5 (0.2) | 7.9 (1.3) | 0.2 (0.0) |
| | EA | 0.6 (0.1) | 1.2 (0.2) | 6.2 (0.7) | 74.4 (14.3) | 2.4 (0.3) | 4.3 (0.9) |
| | ME | 0.4 (0.1) | 1.6 (0.3) | 0.9 (0.4) | 0.1 (0.0) | 54.4 (8.0) | 0.8 (0.2) |
| | RU | 0.6 (0.1) | 10.3 (1.3) | 0.1 (0.0) | 5.1 (2.2) | 4.9 (1.3) | 61.4 (3.1) |
| | Others | 25.6 | 19.7 | 25.8 | 18.6 | 26.4 | 22.5 |
| $NH_x$ Deposition | NA | 88.4 (5.6) | 0.2 (0.1) | 0.3 (0.1) | -* | 0.7 (0.3) | 0.4 (0.2) |
| | EU | 0.6 (0.3) | 83.2 (17.8) | 0.0 (0.0) | - | 4.6 (1.2) | 11.9 (3.1) |
| | SA | 0.0 (0.0) | 0.1 (0.0) | 85.1 (7.6) | - | 8.6 (2.4) | 0.0 (0.0) |
| | EA | 0.0 (0.0) | 0.4 (0.1) | 4.2 (0.3) | - | 2.6 (0.5) | 3.8 (1.0) |
| | ME | 0.1 (0.0) | 1.3 (0.3) | 0.4 (0.2) | - | 49.4 (5.9) | 1.5 (0.4) |
| | RU | 0.4 (0.1) | 10.3 (1.3) | 0.1 (0.0) | - | 7.3 (1.5) | 76.9 (4.1) |
| | Others | 10.5 | 4.4 | 9.7 | - | 26.9 | 5.7 |

* Lack of $NH_4^+$ wet deposition under EA emission perturbation experiment from all models.

Table 2. RERER values of S/$NO_y$/$NH_x$ deposition for continent non-coastal and coastal regions.
Total column gives the RERER for coastal and non-coastal together.

| Regions | S deposition | | | $NO_y$ deposition | | | $NH_x$ deposition | | |
|---|---|---|---|---|---|---|---|---|---|
| | Total | Non-coastal | Coastal | Total | Non-coastal | Coastal | Total | Non-coastal | Coastal |
| NA | 0.17 | 0.12 | 0.40 | 0.17 | 0.12 | 0.43 | 0.07 | 0.05 | 0.31 |
| EU | 0.36 | 0.27 | 0.53 | 0.34 | 0.27 | 0.48 | 0.12 | 0.09 | 0.22 |
| SA | 0.18 | 0.14 | 0.35 | 0.17 | 0.12 | 0.37 | 0.04 | 0.03 | 0.17 |
| EA | 0.16 | 0.14 | 0.28 | 0.16 | 0.12 | 0.27 | - | - | - |
| ME | 0.32 | 0.27 | 0.46 | 0.32 | 0.27 | 0.50 | 0.42 | 0.36 | 0.67 |
| RU | 0.61 | 0.56 | 0.84 | 0.59 | 0.52 | 0.90 | 0.55 | 0.49 | 0.85 |
