# Peer review of "Source contributions to sulfur and nitrogen deposition - an HTAP II multimodel study on hemispheric transport Jiani Tan1, Joshua S. Fu1, Frank Dentener2, Jian Sun1, Louisa Emmons3, Simone Tilmes3, Johannes Flemming4, Toshih"

_Atmospheric Chemistry and Physics, 2018_

## Referee Comment (RC1) · Anonymous Referee #2 · 19 Apr 2018

This study utilizes the HTAP2 perturbation experiments to investigate the source-receptor relationship of the deposition for 6 major world regions. This work was an update study based on the first HTAP study, with more redefined regions and consistent emissions changes. The manuscript is well written and considered to be accepted on ACP after considering the following comments.

I am not convinced by the descriptions of the hemispheric transport on deposition in section 3.2. The range of the fractions (%) seems arbitrary to me and not very illustrative. Please consider alter way to better present the results.

Specific comments:

Pg 3 line 68-69: Since North America and Europe are reused later in the content, suggest to define the abbreviations when they first appeared.

Pg 3 line 76: define the time periods of so-called "increasing trend of the hemisphere transport of air pollution" as well as for the directions of the transport. Since the emissions from NA and Eur have been decreasing for the past decades, the hemispheric transport from these regions to others should be lower. Also lots of studies have shown that Chinese emissions have been decreasing since the peak around 2011 (Liu et al., 2016; Li et al., 2017; Zheng et al., 2017; Zheng et al., 2018).

Pg 3 line 83: Double check Arndt and Carmichael (1995)'s study for the S-R relationship. Is it 1900s, or 1990s?

Pg 3 line 87: change "That study uses" to "That study used"

Pg 3 line 90: change "is deposited" to "was deposited"

Pg 4 line 101: change to "we include 2 more regions"

Pg 4 line 109: remove "other regions"

Pg 4 line 122: extra space in front of the "The project involves"

Pg 5 line 130: I thought the HTAP2 experiments only involved the global CTMs which do not need the BCs?

Pg 5 line 146: remove "are"

Pg 5 line 155: remove the repeat sentence "In terms of wet deposition . . ."

Pg 6 line 161: keep consistent by using either "modelling" or "modeling"

Pg 6 line 174: change "the same emissions" to "constant"

Pg 7 line 203-204: Suggest to change to "less than 3% is deposited"

Pg 7 line 212: From Table 1, the statement is not right. Please clarify. For ME, about

50% deposition in the source region, and more than 70% for the other 5 regions.

Pg 7 line 215: I suspect the authors want to declare that the seasonal variations of S export are around 5-10% for all the regions, except for SA. The sentence was confusing when the author added the annual average of NA which are clearly different from the other regions.

Pg 8 line 228: describe the reasons for the larger export fractions of S/NOx/NH3 in ME

Pg line 233: Should "NOy emissions" be "NOx emissions"?

Pg 10 line 288: change "20" to "20%".

Pg 12 line 360: change "the RERER of NA reached" to "the RERE of NA reaches"

Pg 13 line 378: change the second "own region deposition" to "source region"

Pg 16 line 462-463: rephrase the sentence.

Figure 1. Define region 1.

Figure 7. I have two questions: for the oxidized deposition inter-model comparison (middle plots), the authors should include the organic species (PAN, Orgn), or explain why they did not. For the NH4+ Wet deposition in the bottom plot, I did not see the error bars as other components. Also for NO2 Dry deposition, some regions are missing the error bars too, i.e. EU, ME, RU.

Table 2: Define the RERER in the captions.

In the supporting: Table S3: I would expect to see the seasonal differences for the emission reductions for EA, NA for S and NOx since the HTAP2 emissions have monthly variations (Janssens-Maenhout et al., 2015). Explain why they are always the same amount of reductions.

References: Liu et al., 2016, Recent reduction in NOx emissions over China: synthesis of satellite observations and emission inventories, doi:10.1088/1748-

9326/11/11/114002.

Li et al., 2017, India Is Overtaking China as the World's Largest Emitter of Anthropogenic Sulfur Dioxide, DOI:10.1038/s41598-017-14639-8.

Janssens-Maenhout, G., Crippa, M., Guizzardi, D., Dentener, F., Muntean, M., Pouliot, G., Keating, T., Zhang, Q., Kurokawa, J., Wankmüller, R., Denier van der Gon, H., Kuenen, J. J. P., Klimont, Z., Frost, G., Darras, S., Koffi, B., and Li, M.: HTAP_v2.2: a mosaic of regional and global emission grid maps for 2008 and 2010 to study hemispheric transport of air pollution, Atmos. Chem. Phys., 15, 11411-11432, https://doi.org/10.5194/acp-15-11411-2015, 2015.

Zheng et al., 2017, Air quality improvements and health benefits from China's clean air action since 2013, https://doi.org/10.1088/1748-9326/aa8a32.

Zheng et al., 2018, Rapid decline in carbon monoxide emissions and export from East Asia between years 2005 and 2016, https://doi.org/10.1088/1748-9326/aab2b3.

---

## Referee Comment (RC2) · Anonymous Referee #1 · 1 Jun 2018

This manuscript presents the results from the HTAP II multi-model study on the impact of hemispheric transport to the atmospheric deposition of S and N. Compared to HTAP I studies HTAP II has different definition of regions with higher spatial resolution which provides more accurate information for coastal regions. The manuscript falls within the scope of the journal and is suitable for publication in ACP after a number of corrections that will improve its clarity and are listed below:

1- Explain why the specific regions have been chosen for the study.

2- Title: Source contributions TO sulfur and nitrogen deposition. . .

3- line 34- abstract (but also elsewhere in the text)- west East Asia does not read nicely

4- Line 42: 40% and 23-45% do not add up to 100%, where the remaining at least 15% comes from?

5- Line 68: remove 'to' after '3-5 ppb'

6- Line 77: 'significant'; can you derive a number or better a range of numbers for some pollutants from the publications? to quantify the 'significant' line 87: used

7- Line 160: please clarify whether for the NH4+ wet deposition at stations you compared the modeled NHx deposition with the NH4+ wet deposition measurements, or the NH4+ modeled only. The NH4+ wet deposition measurements are representative of the sum of NH4+ and NH3 deposition.

8- Line 164: The most accurate comparison would have been that between atmospheric observations of gases and aerosols with the modeled concentration levels.

9- Lines 179-180: mass balance requirements. You need to provide more information specific to this study and mention here what tolerance you have for the mass balance. Is it for instance no tolerance at all? 1 per mile ? or higher? This means that you also need to define what means 'almost identical' (line 184).

10- Lines 185-186: Can you explain why the mass balance requirements are not fulfilled for one region while they are fulfilled for the others? What is the particularity of EA?

11- Line 211: please provide a comment on the sources that lead to injections of S emissions higher than NOx emissions in the models. (relevant discussion line 315 – please provide some numbers)

12- Line 225: from the atmosphere

13- Line 226 'summer' do you mean winter?

14- Lines 227-228: indicate the region to which you refer?

15- Line 231: IN our study

16- Lines 236-237: please rephrase in this sentence it is not clear which is what.

17- Line 239: comparable:

18- Lines 267-273: please comment on which circulations patterns contribute to these impacts.

19- Line 275: 'impact similar to that of S emission' does this mean similar lifetime of SOx and NOx ? if yes is this consistent with discussion in lines 211 and 315?

20- Lines 278-288: Could you comment on potential differences in aerosol pH that impact on the NOy partitioning to the aerosol phase?

21- Lines 324-327: briefly describe the reported seasonality.

22- Line 328: replace 'due to that' by 'because'

23- Lines 361-364: can you discuss this based on atmospheric circulation ?

24- Line 381: 'other reasons' & line 388 'joint effects' : can you spell out potential reasons? Could this be change in oxidant chemistry and lifetimes for instance?

25- Line 397: separately

26- Lines 402-404 are very hard to read, please separate and rephrase this sentence for clarity

27- Lines 402-412: I think the units should be Tg(S)/yr or Tg(N)/yr.

28- Lines 427-429: you provide percentages for export of emissions from Europe to Russia and hen from Russia to Europe; similar for EA to RU and RU to EA, why you do not provide the net effect? Please rephrase for clarity.

29- Lines 429-430: remove 'to abroad' since it is export.

30- Line 447: do you mean that ' coastal regions receive upmost half of their deposition

though hemispheric transport from foreign regions' or that 'half of the transported by LRT amount from foreign regions is deposited over the coastal regions' ?

31- Figure captions:

32- Line 745: correct MD to ME

33- Figure 1: why this figure does not show the upper boundaries of regions 3,4,14 and the Arctic?

34- Line 751: add per 0.1x0.1 deg grid box

35- Line 4: According to eq 1 the negative values should indicate increase in deposition with decline in emissions

---

## Author Comment (AC1) · 13 Jul 2018

Attached in the zip file includes responses to referee #1, revised manuscript, and supplementary file.

Please also note the supplement to this comment:
https://www.atmos-chem-phys-discuss.net/acp-2018-109/acp-2018-109-AC1-supplement.zip

---

## Author Response (AR1)

**Referee #1**

**General Comments**

This study utilizes the HTAP2 perturbation experiments to investigate the source receptor relationship of the deposition for 6 major world regions. This work was an update study based on the first HTAP study, with more redefined regions and consistent emissions changes. The manuscript is well written and considered to be accepted on ACP after considering the following comments.

Response: We would like to thank the reviewer for the suggestions to improve the manuscript. Following are the point-by-point responses to the comments.

**Specific comments:**

**Comment**: I am not convinced by the descriptions of the hemispheric transport on deposition in section 3.2. The range of the fractions (%) seems arbitrary to me and not very illustrative. Please consider alter way to better present the results.

Response: we have rewritten the description of the method to calculate the hemispheric transport on deposition in section 3.2.

"Figure 2 is annual response of S deposition to 20% emission reduction in source regions calculated as Eq. (2).

$$Response = \frac{\Delta\, Depo\,(perturbation)}{Depo\,(base)} \times 100\% \qquad (2)$$

where $\Delta$ Depo (perturbation) is the $\Delta$ Depo under perturbation case and Depo (base) is the deposition under base case. The negative values mean that the deposition decreases with reduction in emission."

We replot figure 2, 3 and figure S2 with different color bar. We use a fixed interval in order to make the fraction more revealing. We have also changed the description related to the numbers of fractions in the manuscript.

[Figure]

Fig. 2 The response of S deposition to 20% emission reduction in source regions. The values are the percentage changes (%) in deposition calculated as (changes in deposition with 20% emission reduction) / (base case deposition) ×100%. The unit is % per 0.1×0.1° grid box.

[Figure]

Fig. 3 Same as Fig. 2 but for $NO_y$ deposition. The unit is % per 0.1×0.1° grid box.

**Comment**: Pg 3 line 68-69: Since North America and Europe are reused later in the content, suggest to define the abbreviations when they first appeared.

Response: We have changed it in the manuscript.

**Comment**: Pg 3 line 76: define the time periods of so-called "increasing trend of the hemisphere transport of air pollution" as well as for the directions of the transport. Since the emissions from NA and Eur have been decreasing for the past decades, the hemispheric transport from these regions to others should be lower. Also lots of studies have shown that Chinese emissions have been decreasing since the peak around 2011 (Liu et al.,2016; Li et al., 2017; Zheng et al., 2017; Zheng et al., 2018).

Response: As for the decadal trend of emission, we revise the description in the first paragraph of the Introduction section. Since the modeling time of this study is 2010, the turning point of Chinese emission in 2011 would not affect this study, but is very inspiring for future study.

"The impact is estimated to continue increasing in the near future (Bleeker et al., 2011; Lamarque et al., 2013; Kanakidou et al., 2016; Paulot et al., 2013; Lamarque et al., 2005; Bian et al., 2017). The $NO_x$ emission has increased by about 10 Tg(N) from 2001 to 2010, due to large increase in Asia regions (Tan et al., 2018), but the recent studies reported a turning point for Chinese $NO_x$ emission in 2011 (Li et al., 2017; Liu et al., 2016). On the other hand, the global sulfur (S) emission has declined by about 5 Tg(S) from 2000 to 2010 (Tan et al., 2018). The global fossil fuel $SO_2$ emission has a decreasing trend since 1980 owing to the significant decline in $SO_2$ emission from Europe (EU) and U.S. (Chin et al., 2014). The $SO_2$ emission in China experiences increases from 2000-2005 due to energy consumption and decreases after 2006 thanks to the implementation of Flue-Gas Desulphurization system"

In line 76, we add the time periods for the increasing trend and add the direction of transport in the manuscript.

"Recent studies have reported an increasing trend in the hemispheric transport of air pollution from Asia to NA from mid-1980s to late-2000s. The Asian plume has contributed ~10 ppbv (30%) to the $O_3$ concentration over western NA from mid-1980s to mid-2000s (Jaffe et al., 2003; Parrish et al., 2004), with an annal increase of 0.34-0.50 ppbv $O_3$ (Parrish et al., 2009). More recent study showed the contribution is about 5-7 ppbv $O_3$ in 2006 with an annual increase rate of 1-2 ppb $O_3$ since 2000 (Zhang et al., 2008). The trend well agreed with the rapid growth of Asian emission (Richter et al., 2005; Lu et al., 2010; Verstraeten et al., 2015; Zhang et al., 2007; van der A et al., 2006; van der A et al., 2008)"

**Comment**: Pg 3 line 83: Double check Arndt and Carmichael (1995)'s study for the S-R relationship. Is it 1900s, or 1990s?

Response: It's 1990s. We have changed it.

**Comment**: Pg 3 line 87: change "That study uses" to "That study used"

Response: We have changed it.

**Comment**: Pg 3 line 90: change "is deposited" to "was deposited"

Response: We have changed it.

**Comment**: Pg 4 line 101: change to "we include 2 more regions"

Response: We have changed it.

**Comment**: Pg 4 line 109: remove "other regions"

Response: We have changed it.

**Comment**: Pg 4 line 122: extra space in front of the "The project involves"

Response: We have changed it.

**Comment**: Pg 5 line 130: I thought the HTAP2 experiments only involved the global CTMs which do not need the BCs?

Response: Yes, we deleted "boundary conditions" in the sentence.

**Comment**: Pg 5 line 146: remove "are"

Response: We have changed it.

**Comment**: Pg 5 line 155: remove the repeat sentence "In terms of wet deposition : : :"

Response: We have changed it.

**Comment**: Pg 6 line 161: keep consistent by using either "modelling" or "modeling"

Response: We have changed all the "modelling" to "modeling". We also changed all "modelled" to "modeled".

**Comment**: Pg 6 line 174: change "the same emissions" to "constant"

Response: We have changed it.

**Comment**: Pg 7 line 203-204: Suggest to change to "less than 3% is deposited"

Response: We have changed it.

**Comment**: Pg 7 line 212: From Table 1, the statement is not right. Please clarify. For ME, about 50% deposition in the source region, and more than 70% for the other 5 regions.

Response: Thank you for pointing out the problem. We have revised the sentence to exclude ME. "In terms of NHx deposition, about 20% more $NH_3$ emission is deposited within the source regions (except ME) compared with S and $NO_x$ emission, due to its short lifetime in the atmosphere."

**Comment**: Pg 7 line 215: I suspect the authors want to declare that the seasonal variations of S export are around 5-10% for all the regions, except for SA. The sentence was confusing when the author added the annual average of NA which are clearly different from the other regions.

Response:  Yes, the annual average of 25% is only for NA. In order to make the sentence clear, we delete the 25% in the sentence.

"In terms of S emission, there is 5-10% of seasonal variation around the annual average of export fractions for all regions except SA."

**Comment**: Pg 8 line 228: describe the reasons for the larger export fractions of S/NOx/NH3 in ME.

Response: We added an explanation in line 242-248.

"Therefore, local precipitation washes out large proportion of the air pollutants from the atmosphere during the rainfall seasons. On the other hand, for regions with low precipitation like ME, the percentage of emission removed within own region would be lower than the other regions. In addition, the strong westerly winds in winter and spring favor the hemispheric transport for regions in mid-latitudes of the North Hemisphere. While the rapid vertical convection in summer slows down the zonal transport of air flows and accelerates the local removal process."

**Comment**: Pg line 233: Should "NOy emissions" be "NOx emissions"?

Response: We have changed it.

**Comment**: Pg 10 line 288: change "20" to "20%".

Response: We have changed it.

**Comment**: Pg 12 line 360: change "the RERER of NA reached" to "the RERE of NA reaches"

Response: We have changed it.

**Comment**: Pg 13 line 378: change the second "own region deposition" to "source region"

Response: We have changed it.

**Comment**: Pg 16 line 462-463: rephrase the sentence.

Response: We have rephrased the sentence.

"Meanwhile, there is still a portion of foreign impact that hasn't been attributed in this study (aggregated as other regions "OTH" in Fig. 6). For instance, at least 4 regions (NA, EU, SA and ME) have shown considerable impact (2-10%) on the S and N deposition in North Africa. But since North Africa is not included as a receptor/source region in the perturbation experiments, it is hard to quantify the impact of long-range transport on its deposition. Southeast Asia is regarded as a big emission contributor in Asia. It is important to establish an S-R relationship with other Asian regions. We suggest the future HTAP simulations to include these regions in the perturbation experiments. "

**Comment**: Figure 1. Define region 1.

Response: we have defined region 1 in the caption of figure 1.

"Other regions: **1-Global**, 2-Ocean (including Arctic),"

**Comment**: Figure 7. I have two questions: for the oxidized deposition inter-model comparison (middle plots), the authors should include the organic species (PAN, Orgn), or explain why they did not. For the NH4+Wet deposition in the bottom plot, I did not see the error bars as other components. Also for NO2 Dry deposition, some regions are missing the error bars too, i.e. EU, ME, RU.

Response: We explained the reason in line 400-401: "The figure only shows main compositions of S and N deposition, which together account for more than 95% of total deposition." The amounts of PAN and Orgn are too small and include them in the figure will make the figure more complicated for the reviewers.

Some species do not have error bars, because only 1 model has submitted these species. We added an explanation in the figure caption. "Species without error bars are derived from results of a single model."

**Comment**: Table 2: Define the RERER in the captions.

Response: We have added the definition in the caption.

**Comment**: In the supporting: Table S3: I would expect to see the seasonal differences for the emission reductions for EA, NA for S and NOx since the HTAP2 emissions have monthly variations (Janssens-Maenhout et al., 2015). Explain why they are always the same amount of reductions.

Response:  Thank you for pointing out this problem. We find some problems in the presentation of data. The values are too small with the unit of Tg(S or N) month$^{-1}$. So we change it to $\times$ 0.1 Tg(S or N) month$^{-1}$. We also change the data from seasonal average to monthly average. Following is the updated Table S3 after change.

Table S3. Changes of emission under emission perturbation experiments for 12 months (unit: $\times$ 0.1 Tg(S or N) month$^{-1}$).

| Emission changes | Seasons | Regions of emission perturbation | | | | | |
|---|---|---|---|---|---|---|---|
| | | NA | EU | SA | EA | ME | RU |
| S emission | Jan | -0.913 | -0.646 | -0.860 | -2.575 | -0.505 | -0.463 |
| | Feb | -0.886 | -0.609 | -0.789 | -2.119 | -0.477 | -0.433 |
| | Mar | -0.955 | -0.657 | -0.876 | -2.421 | -0.477 | -0.428 |
| | Apr | -0.942 | -0.591 | -0.822 | -2.173 | -0.452 | -0.395 |
| | May | -0.956 | -0.483 | -0.840 | -2.188 | -0.436 | -0.360 |
| | Jun | -0.996 | -0.476 | -0.800 | -2.249 | -0.415 | -0.332 |
| | Jul | -1.009 | -0.449 | -0.805 | -2.231 | -0.411 | -0.323 |
| | Aug | -1.007 | -0.392 | -0.799 | -2.184 | -0.425 | -0.322 |
| | Sep | -0.961 | -0.452 | -0.792 | -2.168 | -0.423 | -0.343 |
| | Oct | -0.971 | -0.512 | -0.834 | -2.234 | -0.459 | -0.385 |
| | Nov | -0.956 | -0.514 | -0.826 | -2.521 | -0.472 | -0.398 |
| | Dec | -0.911 | -0.602 | -0.877 | -2.820 | -0.496 | -0.443 |

| Emission changes | Seasons | Regions of emission perturbation | | | | | |
|---|---|---|---|---|---|---|---|
| | | NA | EU | SA | EA | ME | RU |
| NOx emission | Jan | -0.698 | -0.424 | -0.536 | -1.434 | -0.228 | -0.236 |
| | Feb | -0.697 | -0.427 | -0.484 | -1.249 | -0.228 | -0.231 |
| | Mar | -0.730 | -0.462 | -0.533 | -1.435 | -0.227 | -0.227 |
| | Apr | -0.729 | -0.446 | -0.512 | -1.362 | -0.227 | -0.219 |
| | May | -0.728 | -0.407 | -0.525 | -1.369 | -0.221 | -0.202 |
| | Jun | -0.767 | -0.408 | -0.505 | -1.400 | -0.217 | -0.195 |
| | Jul | -0.768 | -0.388 | -0.516 | -1.399 | -0.210 | -0.186 |
| | Aug | -0.768 | -0.366 | -0.514 | -1.389 | -0.215 | -0.191 |
| | Sep | -0.730 | -0.391 | -0.492 | -1.365 | -0.215 | -0.194 |
| | Oct | -0.731 | -0.424 | -0.527 | -1.374 | -0.229 | -0.217 |
| | Nov | -0.730 | -0.415 | -0.505 | -1.494 | -0.231 | -0.224 |
| | Dec | -0.698 | -0.424 | -0.537 | -1.587 | -0.229 | -0.232 |

| Emission changes | Seasons | Regions of emission perturbation | | | | | |
|---|---|---|---|---|---|---|---|
| | | NA | EU | SA | EA | ME | RU |
| NH3 emission | Jan | -0.391 | -0.423 | -2.102 | -1.122 | -0.080 | -0.130 |
| | Feb | -0.437 | -0.487 | -1.805 | -1.026 | -0.146 | -0.235 |
| | Mar | -0.591 | -0.780 | -2.098 | -1.205 | -0.246 | -0.412 |
| | Apr | -0.694 | -0.850 | -1.996 | -1.487 | -0.187 | -0.338 |
| | May | -0.719 | -0.742 | -2.097 | -1.765 | -0.117 | -0.224 |
| | Jun | -0.933 | -0.627 | -1.996 | -1.920 | -0.110 | -0.200 |
| | Jul | -1.077 | -0.571 | -2.096 | -1.904 | -0.113 | -0.199 |
| | Aug | -0.903 | -0.574 | -2.096 | -2.038 | -0.126 | -0.215 |
| | Sep | -0.676 | -0.606 | -1.996 | -1.531 | -0.121 | -0.214 |
| | Oct | -0.632 | -0.642 | -2.097 | -1.312 | -0.094 | -0.178 |
| | Nov | -0.592 | -0.617 | -1.998 | -1.360 | -0.074 | -0.146 |
| | Dec | -0.344 | -0.517 | -2.109 | -1.263 | -0.074 | -0.133 |

After the update, we compare our results with the emission inventory from Janssens-Maenhout et al., (2015). In the figure 1(c) of Janssens-Maenhout et al., (2015) as shown in below, the US and CANADA emissions have large monthly variations in agriculture sector, but are relatively steady in other emission sectors.

[Figure]

Figure 1(c) from Janssens-Maenhout et al., (2015). Temporal profiles with relative factors varying around 1/12 and applied on the yearly emissions of the different data sources (US-EPA for US and Canada, EMEP-TNO for Europe with compound-specific variation of the agricultural temporal profiles, EDGAR temporal profiles for the Northern Hemisphere and MICS profiles for Asia).

Here we plot the amounts of emission changes with 20% emission reduction in this study for $SO_2$ (mainly from energy and industry sectors), $NO_x$ (mainly from energy, industry sources and transport sectors) and $NH_3$ (mainly from agriculture sector). The reduction of $NH_3$ is largest in July, while the reduction of $SO_2$ and $NO_x$ are relatively steady throughout the year, which is consistent with the changing trend of emission in US and CANADA.

Changes of emission under NA case for 12 months (unit: $\times$ 0.1 Tg(S or N) month$^{-1}$)

[Figure]

Following figure is the characteristic of monthly trends of European emissions from Janssens-Maenhout et al., (2015). The European emission has large monthly change in energy and residential sectors, which are high in March and low in August. While the agriculture emission is highest in March and April.

[Figure]

Figure 1(c) from Janssens-Maenhout et al., (2015). Temporal profiles with relative factors varying around 1/12 and applied on the yearly emissions of the different data sources (US-EPA for US and Canada, EMEP-TNO for Europe with compound-specific variation of the agricultural temporal profiles, EDGAR temporal profiles for the Northern Hemisphere and MICS profiles for Asia).

We plot the mounts of emission changes with 20% emission reduction in this study for S, $NO_x$ and $NH_3$ emission in Europe. S and $NO_x$ emissions have largest reductions in March and smallest reduction in August, consistent with the trend of energy and residential sectors. The European $NH_3$ emission has the largest reduction in March and April, also agrees well with the variation of agriculture emission.

Changes of emission under EU case for 12 months (unit: $\times$ 0.1 Tg(S or N) month$^{-1}$)

[Figure]

7- Line 160: please clarify whether for the NH4+ wet deposition at stations you compared the modeled NHx deposition with the NH4+ wet deposition measurements, or the NH4+ modeled only. The NH4+ wet deposition measurements are representative of the sum of NH4+ and NH3 deposition.

Response: We have added the following sentences to clarify.

"Modeled gas phase $SO_2$ wet deposition and aerosol $SO_4^{2-}$ wet deposition are evaluated with observed $SO_4^{2-}$ wet deposition."

"Modeled gas phase $HNO_3$ wet deposition and aerosol $NO_3^-$ wet deposition are compared with observed $NO_3^-$ wet deposition."

"Modeled gas phase $NH_3$ wet deposition and aerosol $NH_4^+$ wet deposition are compared with observed $NH_4^+$ deposition."

8- Line 164: The most accurate comparison would have been that between atmospheric observations of gases and aerosols with the modeled concentration levels.

Response: We have compared both air concentration and dry deposition velocity between CASTNET and multi-model results. The results are shown in our previous paper "Multi-model study of HTAP II on sulfur and nitrogen deposition". Following are the related paragraphs in that paper for reference.

"Since the CASTNET dry deposition is not actually measured but instead a calculation of measured concentration of species and modelled dry deposition velocities, it is necessary to investigate which factor of these two contributes to the model bias. We compare the modelled air pollutant concentrations with CASENET measurements as shown in Table S4-S8. The MMM overestimates the $SO_2$, $SO_4^{2-}$, $HNO_3$, $NO_3^-$ and $NH_4^+$ concentrations by 394%, 40%, 217%, 135%

and 173%, respectively. It should be noted that the CASTNET sites are generally located in rural regions that are away from emission sources (Sickles and Shadwick, 2008), thus the measured concentrations of air pollutants are relatively low compared with those of urban sites. While the resolutions of the HTAP II models range from 0.5° to 3°, and are not fine enough to reproduce the characteristic of some rural sites. The models with finer resolutions except CHASER_t106 model (i.e. EMEP_rv48 (0.5° × 0.5°) and SPRINTARS (1.1° × 1.1°)) generally perform better than the others, while models with coarse resolutions (i.e. CHASER_re1 (2.8° × 2.8°) and OsloCTM3.v2 (2.8° × 2.8°)) are generally not performing well for all species. This could explain the overestimation of air pollutant concentrations at the CASTNET sites.

In order to check the differences of modelled dry deposition velocity between CASNET and HTAP II models, we adopt the general approach for calculating dry deposition velocity from Wesely, (1989). $V_d = -F_c / C_a$ (7), where $V_d$ is the deposition velocity, $F_c$ is the dry deposition flux and $C_a$ is the concentration of species. The negative mark indicates the direction of the dry deposition velocity. This scheme has been widely adopted in global models (Wesely and Hicks, 2000) with modifications. We compare the calculated dry deposition velocity of models and CASTNET (Table S9-S13). The mean bias of dry deposition velocities for MMM are -8%, 0.3%, 7%, 19% and 2% for $SO_2$, $SO_4^{2-}$, $HNO_3$, $NO_3^-$ and $NH_4^+$, respectively, which are much lower than those of air pollutants. The model bias for dry deposition at the CASTNET sites mainly comes from the model over prediction of air pollutant concentration."

To make it clear, we add short description in this manuscript as follows:

"The CASTNET data is calculated with observed aerosol concentration and modeled dry deposition velocity, therefore it has high uncertainty in data quality. Evaluation shows that the modeled dry deposition is generally higher than the CASTNET inferential data by a factor of 1-2. This is a common feature of many global and regional models (WMO, 2017). According to the analysis, the model bias for dry deposition at the CASTNET sites mainly comes from the model over-prediction of air pollutant concentration. The CASTNET sites are generally located in remote regions with relatively lower air pollutant concentrations than urban regions, but the models fail to represent this characteristic with coarse spatial resolution (Tan et al., 2018)."

9- Lines 179-180: mass balance requirements. You need to provide more information specific to this study and mention here what tolerance you have for the mass balance. Is it for instance no tolerance at all? 1 per mile ? or higher? This means that you also need to define what means 'almost identical' (line 184).

Response: We added the criteria in the manuscript.

"We compare the global total amounts of changes of deposition (Δ Depo) with changes of emissions (Δ Emis) for all perturbation cases (Table S1). Models are excluded if their global Δ Depo values fall outside the range of ±20% of their global Δ Emis."

10- Lines 185-186: Can you explain why the mass balance requirements are not fulfilled for one region while they are fulfilled for the others? What is the particularity of EA?

Response: Sorry for the confusion. The change of deposition under EA case is missing because no model has submitted the wet deposition of $NH_4^+$ under the EA perturbation case, although they have submitted the data under base case. We have rephrased the sentence in the manuscript.

"The Δ Depo of $NH_x$ deposition under EA case is not available due to lack of model results for the wet deposition of $NH_4^+$ under 20% emission perturbation in EA."

11- Line 211: please provide a comment on the sources that lead to injections of S emissions higher than NOx emissions in the models. (relevant discussion line 315 – please provide some numbers)

Response: The injection heights of S and NOx emission are different in the emission inventory due to their emission sources. The following figure shows the contribution of emission sectors to air pollutants in the emission inventory of HTAP II (Janssens-Maenhout et al., 2015). $SO_2$ mainly comes from power plants and industry (~80%). Both of these two sectors have high outlets. For $NO_x$, the contribution of these two sectors are much lower (~40%). About 40% of $NO_x$ is from transport section, which emits pollutant near the earth surface. Thus, we conclude that the S emissions are emitted from higher altitude than $NO_x$ emission. However, we notice that some global models (such as CAM-Chem and C-IFS_v2 models) treat power plant emissions at first layer. In that case, the injection height won't affect the results, but the generally longer lifetime of S emission could affect the results.

[Figure]

Figure from (Janssens-Maenhout et al., 2015) Global sector-specific anthropogenic emissions of gaseous pollutants and particulate matter components for the year 2010. Global absolute emissions are reported on top of each bar in Tg species per year. Large scale open-biomass burning is not included in the analysis.

12- Line 225: from the atmosphere

Response: We have changed it.

13- Line 226 'summer' do you mean winter?

Response: We notice that summer or winter is not the rainfall season for all regions. We deleted it in the manuscript.

14- Lines 227-228: indicate the region to which you refer?

Response: We rewrite this sentence for clarification.

"In addition, the strong westerly winds in winter and spring favor the hemispheric transport for regions in mid-latitudes of the North Hemisphere. While the rapid vertical convection in summer slows down the zonal transport of air flows and accelerates the local removal process."

15- Line 231: IN our study

Response: We have changed it.

16- Lines 236-237: please rephrase in this sentence it is not clear which is what.

Response: We have rephrased the sentence.

"HTAP I study by (Sanderson et al., 2008) developed a SR relationship for $NO_y$ deposition among NA, EU, SA and EA. Their results showed that about 12-24% of the emitted $NO_x$ is deposited out of source regions. This study of HTAP II finds a higher percentage of export (26-34%)."

17- Line 239: comparable:

Response: We have changed it.

18- Lines 267-273: please comment on which circulations patterns contribute to these impacts.

Response: We find the circulation pattern is strongly related to the seasonality of long-range transport. So we described the circulation with seasonality in another paragraph in line 340-356.

"The deposition change via transport between neighboring regions is found throughout the whole year and is slightly stronger in winter, such as between EU and RU (~30°E) (Fig. 4(b) and (f)) and from EA to the North Pacific Ocean (~130°E) (Fig. 4(d)). This is consistent with the seasonality we found for the export of emission by source regions in section 3.1. In addition, most source regions reduce more S and $NO_x$ emissions in winter than the other seasons (Table S3), thus more emissions are exported abroad in winter. On the contrary, most of the deposition change by transport over long distance occurs in spring and fall, especially for the hemispheric transport from NA to EU, from EU to EA and from EA to NA. The seasonality of long-range transport for NA, EU and EA well fits the characteristic of westerlies, which is the prevailing winds in the mid-latitude of the North Hemisphere. This agrees with the seasonality of the transpacific, transatlantic and trans-Eurasia flows of air pollutants that spring is the most efficient season for long-range transport for mid-latitudes. (Holzer et al., 2005;Liu et al., 2005;Liang et al., 2004;Brown-Steiner and Hess, 2011;Li et al., 2014;Auvray and Bey, 2005;Wild et al., 2004;Liu et al., 2003). Although the westerlies is also strong in winter, the long distance transport of emissions is low, because the formation of secondary species like PAN is suppressed by slow oxidation in cold environment (Berntsen et al., 1999;Deolal et al., 2013;Moxim et al., 1996), which plays an important role as a reservoir for $NO_x$ in the long-range transport of air plumes (Lin et al., 2010;Hudman et al., 2004)."

19- Line 275: 'impact similar to that of S emission' does this mean similar lifetime of SOx and NOx ? if yes is this consistent with discussion in lines 211 and 315?

Response: No, S has longer lifetime than $NO_x$. We rewrite the sentence.

"The overall impact is qualitatively similar to that of S emission in the spatial pattern, with some differences in the values."

20- Lines 278-288: Could you comment on potential differences in aerosol pH that impact on the NOy partitioning to the aerosol phase?

Response: This is an interesting question. The pH value affects the dissolution of gas-phase $HNO_3$. The gas-phase $HNO_3$ is produced by daytime reaction Eq. (1), and is very soluble in water (Eq. (2)) and dissociates to aerosol phase $NO_3^-$ (Eq. (3)). Under the ideal solution, the higher the pH value, the more gas-phase $HNO_3$ can be dissolved.

$$NO_2 + OH \rightarrow HNO_3 \text{ (gas)} \qquad (1)$$
$$HNO_3 \text{ (g)} \rightleftharpoons HNO_3 \text{ (aq)} \qquad (2)$$
$$HNO_3 \text{ (aq)} \rightleftharpoons NO_3^- + H^+ \qquad (3)$$

In this case, the perturbation in $SO_2$ emission decreases the concentration of acid $SO_4^{2-}$ in aerosol, which increases the aerosol pH value. If ignore the effects on the dissolution of $NH_3$, this condition will increase the fraction of $NO_y$ partitioned to aerosol phase. In addition, study by (Bian et al., 2017) found that the $NH_3$ wet deposition is very sensitive to the pH value in the cloud, which as a result will largely affect the $NH_3$-$NH_4^+$-$NO_3^-$ equilibrium. The models without pH adjustment can provide very different results of $NO_3^-$ aerosol with models that with the adjustment.

21- Lines 324-327: briefly describe the reported seasonality.

Response: We have changed in the manuscript.

"This agrees with the seasonality of the transpacific, transatlantic and trans-Eurasia flows of air pollutants that spring is the most efficient season for long-range transport for mid-latitudes."

22- Line 328: replace 'due to that' by 'because'

Response: We have changed it.

23- Lines 361-364: can you discuss this based on atmospheric circulation?

Response: We have added the following description for circulation.

"Except large scale circulation like prevailing westerlies, the coastal regions are featured with complex small scale circulations. For instance, the low-level jet (zonal winds with high speed) contributes to the rainfall in coastal regions in Asia (Xavier et al., 2018). The orographic effects enhance the precipitation over coastal mountain regions such as west coast of NA, EU and southeast coast of RU (James and Houze, 2005)."

24- Line 381: 'other reasons' & line 388 'joint effects' : can you spell out potential reasons? Could this be change in oxidant chemistry and lifetimes for instance?

Response: we explain the "other reason" in following sentence:

"It indicates the deposition change due to other reasons than the total effects of separate emission reduction in the 6 regions. It could come from the emission reduction in rest of world, especially nearby regions such as from Central Asia and North Africa to EU and ME. It could also come from the joint effects of emission control by multiple source regions, which possibly change the oxidant chemistry, atmospheric mixing and lifetimes of reactive pollutants. However, the model simulations do not allow to separate these two contributions in this study."

25- Line 397: separately

Response: We have changed it.

26- Lines 402-404 are very hard to read, please separate and rephrase this sentence for clarity

Response: we have rephrased the sentence.

"In terms of S deposition (Fig. 7(a)), the ranges of modelled $\Delta$ Depo by multiple models, defined as (modelled maximum value – modelled minimum value), are (0.06-0.23) Tg(N) yr$^{-1}$ and (0.01-0.22) Tg(N) yr$^{-1}$ for $SO_2$ dry and wet deposition, and (0.01-0.03) Tg(N) yr$^{-1}$ and (0.009-0.17) Tg(N) yr$^{-1}$ for $SO_4^{2-}$ dry and wet deposition, respectively."

We also change the corresponding sentences for $NO_y$ and $NH_x$ deposition.

"In terms of $NO_y$ deposition (Fig. 5(b)), the differences among models range (0.003-0.07) Tg(N) yr$^{-1}$ for $NO_2$ dry deposition, and (0.07-0.55) Tg(N) yr$^{-1}$ and (0.03-0.75) Tg(N) yr$^{-1}$ for $NO_3^-$ dry and wet deposition, respectively."

"In terms of $NH_x$ deposition (Fig. 5(c)), the differences among models range (0.04-0.09) Tg(N) yr$^{-1}$ for $NH_3$ dry deposition, and (0.008-0.15) Tg(N) yr$^{-1}$ and (0.002-0.11) Tg(N) yr$^{-1}$ for $NH_4^+$ dry and wet deposition, respectively."

27- Lines 402-412: I think the units should be Tg(S)/yr or Tg(N)/yr.

Response: We have changed it.

28- Lines 427-429: you provide percentages for export of emissions from Europe to Russia and hen from Russia to Europe; similar for EA to RU and RU to EA, why you do not provide the net effect? Please rephrase for clarity.

Response: The export fraction are related to the total emission amount and seasonal variation of the source regions, thus we prefer to describe the fraction separately for each region. In order to make it clear, we rewrite the sentence:

"The most significant exports of emissions are: 1) transport between EU and RU. 10-14% of EU's emission is transported to RU and 7-12% of RU's emission is transport to EU. 2) transport between EA and RU. 5% of EA's emission is transported to RU and 4-5% of RU's emission is transported to EA. 3) transport from SA to EA (4-9%)."

29- Lines 429-430: remove 'to abroad' since it is export.

Response: We have changed it.

30- Line 447: do you mean that 'coastal regions receive upmost half of their deposition though hemispheric transport from foreign regions' or that 'half of the transported by LRT amount from foreign regions is deposited over the coastal regions' ?

Response: The later one is the appropriate understanding. We rephrase the sentence in the manuscript.

"For some regions, upmost half of the hemispheric transport from foreign regions is deposited over their coastal regions."

31- Figure captions:

Response: We have changed it.

32- Line 745: correct MD to ME

Response: We have changed it.

33- Figure 1: why this figure does not show the upper boundaries of regions 3,4,14 and the Arctic?

Response: We re-do the figure for clarity. The upper boundaries of regions 3,4 and 14 are region 16-Arctic.

[Figure]

34- Line 751: add per 0.1x0.1 deg grid box

Response: We have changed it. We also change the unit in the caption of figure 3.

35- Line 4: According to eq 1 the negative values should indicate increase in deposition with decline in emissions

Response: Yes, thus table 1 are all possible values, which means decrease in emission lead to decrease in deposition.

Reference:

[revised manuscript text omitted]